# Improving Data-Informed Care in New Brunswick Long-Term Care Homes: A Qualitative Study on an Educational Intervention for interRAI Coordinators

**DOI:** 10.3390/healthcare12242592

**Published:** 2024-12-23

**Authors:** Rachel MacLean, Pamela Durepos, Lisa Keeping-Burke, Rose McCloskey

**Affiliations:** 1Department of Interdisciplinary Studies, University of New Brunswick, 3 Bailey Drive, Fredericton, NB E3B 5A3, Canada; 2Faculty of Nursing, University of New Brunswick, 33 Dineen Drive, Fredericton, NB E3B 3X9, Canada; 3Department of Nursing and Health Sciences, University of New Brunswick, 100 Tucker Park Rd, Saint John, NB E2K 5E2, Canada; lkeeping@unb.ca (L.K.-B.); rmcclosk@unb.ca (R.M.); 4The University of New Brunswick (UNB) Saint John Collaboration for Evidence-Informed Healthcare: A JBI Centre of Excellence, University of New Brunswick, 100 Tucker Park Rd, Saint John, NB E2K 5E2, Canada

**Keywords:** educational intervention, interRAI, long-term care

## Abstract

**Background/Objectives**: InterRAI is a globally validated platform aimed at improving care for individuals with disabilities and complex medical needs, particularly in long-term care settings. This study explores the experiences of interRAI coordinators in New Brunswick, Canada, and their perceptions of an educational intervention designed to enhance their ability to effectively use interRAI data for quality care. **Methods**: The study recruited interRAI coordinators from 73 New Brunswick long-term care homes for an educational intervention. Nine coordinators participated in interviews about their experiences. A qualitative descriptive approach was used to analyze field notes and interview transcripts with thematic analysis. **Results**: Nine interviews and six sets of field notes were collected over one year, focusing on the roles of interRAI coordinators. Participants (all female, averaging 54 years old) expressed positive perceptions of the intervention, noting increased knowledge and collaboration. Key themes included the context of the interRAI coordinator role, the use of interRAI data for quality indicators, and recommendations for future educational initiatives. **Conclusions**: The findings emphasize the critical role of interRAI coordinators in improving quality care in long-term care settings through effective data use and collaboration. Participants reported that the educational intervention significantly improved their understanding and application of interRAI data. Recommendations for ongoing training and broader engagement stress the importance of continuous support to advance care quality in long-term care homes.

## 1. Introduction

The International Resident Assessment Instrument (interRAI) is a validated, evidence-based platform utilized in over 35 countries to gather, analyze, and apply data to enhance the care of individuals with disabilities or complex health needs [1]. The interRAI system is employed in various settings, including acute care in Australia [2], home care in Iceland [3], and long-term care (LTC) in Canada. In LTC homes, data are gathered by staff through direct observation, medical records, and communication among residents, families, and the healthcare team. Data are input into interRAI where an interRAI coordinator oversees them and can utilize the data to lead initiatives and QI (quality indicator) projects, guide individualized care planning, track care quality across facilities and against national standards, and identify improvement needs [4]. As such, interRAI data are instrumental in enhancing residents’ quality of life and care; however, there is a significant gap in understanding interRAI coordinators’ capacity to fully leverage the system and use interRAI data within their homes, hampering the ability to optimize their impact on care quality. This study seeks to explore the experiences of interRAI coordinators in New Brunswick (NB) LTC homes and their perceptions of an educational intervention aimed at improving their capacity to use interRAI data in delivering evidence-informed care.

The challenges faced by the LTC sector globally, such as the rapidly aging population and increasingly complex health needs among older adults, raise serious concerns about the sector’s capacity to provide adequate care in the face of growing demand [5]. In New Brunswick, for instance, the vast majority of older adults report managing one or more chronic conditions (85.9%), living with disabilities (35.6%), and experiencing unmet mental health needs (30.1%) [6]. The interRAI system offers a valuable solution for caring for high volumes of older adults with varying care needs by enabling the systematic measurement, tracking, and reporting of Quality Indicators (QIs) within LTC settings. This approach is crucial for promoting care transparency, identifying areas of concern, establishing global standards for patient-centered care, and guiding effective, evidence-based care decisions for this vulnerable population [5,7].

The interRAI system encompasses a range of features designed to inform person-centered care planning and population-level research. One feature is clinical assessments, which use various scales to gather data on factors such as social engagement, functional status, and quality of life [8,9], and are useful for guiding care plans and are a valuable resource for researchers. For example, Canadian researchers utilized interRAI data to identify modifiable risk factors for cognitive decline in older adults [10], while researchers in New Zealand applied the system’s data to inform policies regarding Indigenous data sovereignty [11]. Although the potential of interRAI data to guide clinical practice and policy is clearly significant, it can only do so if LTC staff are well-equipped to use the system effectively.

Research has shown that the effective use of interRAI data can significantly improve care quality in LTC settings. For instance, a recent observational study found a strong correlation between using interRAI data and improved influenza vaccination rates across several countries, highlighting its impact on care outcomes in LTC homes [12]. Additionally, a review by Osinska et al. [7] emphasized the importance of utilizing interRAI QIs to address key issues such as falls, pressure ulcers, and physical restraints in LTC. In Canada, interRAI data have been crucial in assessing the impact of COVID-19 lockdowns on residents’ mental health, offering valuable insights for developing strategies to mitigate these effects [9]. Additionally, research has been conducted on interRAI communities of practice, such as the Seniors Quality Leap Initiative (SQLI) in Canada and the Les Visionnaires network in New Brunswick, harnessing the power of interRAI in enhancing care quality across LTC homes. Through these communities, LTC homes have been able to compare their performance on QIs and exchange strategies for improvement [8]. As a result of the SQLI community of practice, participating homes achieved substantial reductions in restraint use, with some homes reporting decreases of up to 91% [13]. Despite the support for using LTC interRAI data, there remains a lack of comprehensive understanding regarding coordinators’ experiences with and perceptions of the data. Given that many jurisdictions mandate the collection and reporting of interRAI data, it is critical to understand how interRAI coordinators use the system to inform their care practices.

To address these gaps, Canada’s national interRAI organization implemented an educational intervention across NB, including a series of virtual sessions and open-call meetings. This initiative, which ran from January to December 2023, aimed to build capacity among LTC staff and foster peer support for using interRAI data to improve care. Through structured education on systematic data collection, data interpretation, and quality improvement strategies, along with informal discussions among LTC staff, the initiative sought to enhance the application of interRAI data in day-to-day care. The feedback from interRAI coordinators participating in this intervention will provide valuable insights into the effectiveness of the training and the practical challenges of using interRAI data in NB LTC homes.

This study will thus contribute to the broader understanding of how interRAI coordinators experience and utilize interRAI data and whether educational interventions can help address any barriers to its full use. By exploring these issues, the study will inform future strategies to improve the quality of care in LTC settings and help build a more sustainable, evidence-informed care model for aging populations.

### Research Questions

How do participants in the educational intervention engage with and react to the content and each other, including ways to work toward a community of practice? What are interRAI coordinators’ experiences and perceptions of the educational intervention for building capacity and peer support in LTC homes? What is the context of their role and how are the data collected through interRAI being utilized in NB LTC? 

## 2. Materials and Methods

### 2.1. Educational Intervention

Over a 12-month period from January to December 2023, a series of four structured virtual education sessions were conducted by an academic leader from the national interRAI organization. Additionally, nine virtual ‘open-call’ meetings were held immediately following and between the educational sessions. These sessions included simultaneous translation from English to French to accommodate staff from Anglophone and Francophone LTC homes across the province. Open-call meetings were facilitated by a leader from the provincial LTC home association, which operates on a non-governance, voluntary membership basis. Participation in these interventions was entirely voluntary. The educational sessions featured a one-hour presentation on systematic data collection methods (including variable definitions), data interpretation, comparisons to national benchmarks, and strategies for using interRAI data to inform quality improvement. Open-call meetings provided a platform for open dialogue, questions, and peer-to-peer sharing about experiences and concerns related to interRAI. Some meetings included presentations from guest speakers, such as LTC staff, who shared quality improvement projects informed by interRAI data. The goals of the intervention were two-fold: to build capacity for collecting and using interRAI data and to foster peer support, creating a community of practice among LTC homes to enhance the application of interRAI data for quality improvement.

### 2.2. Participant Recruitment and Sampling

This study was approved by the University of New Brunswick Research Ethics Board #2022-114. Invitations to participate in the educational intervention were sent to Anglophone and Francophone LTC staff responsible for collecting and reporting interRAI data from New Brunswick’s 73 LTC homes through email invitations to the LTC home administrator or leader. Prior to the first education session, administrators/leaders were emailed a study invitation and participant information sheet and were requested to share this email with the most appropriate person in their facility. The message explained researchers would be present during the intervention (i.e., education sessions and open calls) to observe and take field notes that would be anonymized and aggregated, and that individuals may also be invited to participate in an interview with a research assistant. This information was reviewed at each session to ensure participants were aware of the researcher’s presence.

For the education/open-call sessions, implied consent was obtained using the text ‘chat’ feature of ZOOM software (version 5.17.11) during the virtual education and meetings. For the one-on-one interviews, informed consent was collected prior to participating in an interview. Participants could request their data (e.g., comments, interview transcript) be excluded from the study or withdrawn at any time by emailing the researchers. When 75% of the intervention (i.e., 3/4 education sessions and open calls) had been completed in Fall 2023, participants who had actively engaged in at least one of the education or open-call sessions (either by making oral or written comments) were invited to participate in a virtual interview with a research assistant by email.

### 2.3. Approach

To ensure transparency and rigor in our study, we adhered to the SRQR reporting guidelines for qualitative studies (see Appendix A). This study utilized a qualitative descriptive approach to produce a detailed description of the phenomenon [14] through an interpretivist paradigm focused on interpreting the meaning and context of observed behaviors, experiences, or interactions, acknowledging the subjectivity and context of the research [15]. Attendance at the education and open-call sessions was tracked. One to three researchers attended each education session and open-call meeting and independently collected data using a standardized field note template. Categories on the template aimed to capture attendees’ experiences, perceptions, questions, reactions, and uses of the interRAI data. Education sessions and open calls were recorded to allow attendees to access the information; however, the recordings were not made available to the researchers as a data source, supporting anonymity and avoiding the need to collect informed consent from 100+ fluctuating participants at the sessions.

Virtual interviews were conducted through Microsoft Teams using a semi-structured interview guide with questions such as: “What was surprising about the interRAI data in the sessions?” and “What quality improvements has your home initiated based on interRAI data?” (the full interview guide can be found in the Appendix A). The interviewer was a nurse external to the study with experience in conducting qualitative interviews. Interviews were recorded and transcribed by the interviewer and proofed against the audio by a research assistant (RA). The participants were unknown to the researchers and the researchers did not interact with the participants.

### 2.4. Analysis

Field notes and transcripts were uploaded into Dedoose (Sociocultural Consultants, Inc., Los Angeles, California, USA), a qualitative data analysis platform. Two RAs independently analyzed three transcripts using thematic analysis and then met for research triangulation to confirm codes and emerging themes. They agreed on a coding scheme through reflexive discussion, completed coding the remaining transcripts and field notes [16], and regularly met with the larger research team to share progress. To provide different perspectives on the same research questions, descriptive summaries of the field notes and interviews are presented separately, supporting data triangulation and reducing bias.

## 3. Results

An average of 51 LTC staff participants attended the education sessions and open-call meetings. Nine interviews lasting 45 min on average were completed and six field notes were collected during the education intervention sessions over one year (2022–2023). 

### 3.1. Participant Characteristics

All nine interview participants identified as female and were interRAI coordinators (100%). The description of the participants only reflects data collected from six participants as three did not return the demographic surveys (see Table 1). Amongst these, participants averaged 54 years of age (SD 12.09). Two participants worked as both interRAI coordinators and staff nurses, and the majority were part-time employees (66.7%). Participants had worked in LTC for 19.1 years on average (SD 17.56) and had been in the role of interRAI Coordinator for 5.8 years (SD 2.3). Participants had attended 75% of the education sessions on average (Median 3, SD 1.51; Range = 6–10) and 39% of the open-call meetings (Median 3.5, SD 2.37). In terms of intervention adherence, four interviewed participants had attended all of the education sessions and open-call meetings, two had attended most (>50%), and three had attended few (<49%) sessions and meetings.

### 3.2. Field Note Narrative Summary

The education sessions and open-call meetings were characterized by engagement, learning, and mutual support, with participants motivated to share ideas and collaborate. While some sessions began with uncertainty, they shifted toward increased energy and enthusiasm as they progressed. The leadership tone was positive, using humor and motivation to foster a non-punitive learning environment. Support and networking were emphasized, with participants exchanging experiences and offering practical solutions.

Discussion of interRAI data and QIs was central to the sessions. Antipsychotic usage, particularly reducing inappropriate use and addressing coding challenges, was frequently discussed. Restraint use was another major focus, with NB’s higher-than-average restraint rates prompting discussions on reducing restraints through policies and innovations like buckling seatbelts behind wheelchairs (field note 3). Fall rates also sparked debate, particularly around definitions and accurate tracking. A consistent takeaway from discussions surrounding interRAI data was to engage with and learn from LTC homes with better QI scores.

Province-wide interRAI data were shared, prompting discussion for clarifying definitions, such as whether tilt chairs count as restraints (field note 2), and discussions about facility funding and policies as barriers to implementing alternative approaches to QIs, such as resource allocation impacting restraint reduction. Participants shared strategies, including policies (field note 3) and family education methods (field note 4). Many noted the need for a culture shift, with one participant stating, “It is a real culture shift for some to move away from things like gerichairs!!! Lots of resistance from staff and families. Crazy!” (field note 1).

Participants highlighted strengths and concerns in using interRAI data to improve care. Strengths included a positive culture shift with more engagement and shared goals, and successful reductions in QIs like antipsychotic use (field note 2). Concerns included resistance from families and physicians, particularly regarding restraint reduction, culture, and policy barriers like funding limitations, and challenges in ensuring accurate data entry.

The sessions also involved extensive discussions on strategies for improvement, such as reducing bed alarm use, prioritizing injury prevention over fall prevention, sharing policies across homes, advocating for recreational engagement, and involving families and staff in QI education. Participants outlined plans for utilizing interRAI data, including establishing protocols for consistent data management, holding monthly coordinator meetings, and implementing behavior mapping plans to address resident behavior comprehensively.

### 3.3. Interviews Narrative Summary

A narrative summary was generated through thematic analysis providing insight into the context of the interRAI coordinator role, including their responsibilities, activities, and previous education and training. They discussed how interRAI data are used, the quality improvement strategies employed, and shared mainly positive perceptions of the intervention. Facilitators and barriers to implementation were identified, as well as perceived benefits like increased knowledge, collaboration, and improved practice. Participants also offered recommendations for further education and training.

#### 3.3.1. Theme 1: Context

Participants provided an overview of the role of the interRAI coordinator, staff involvement with interRAticipants prI data, and processes for collecting, interpreting, utilizing, monitoring, and communicating about quality indicators in their respective homes. The interRAI coordinator role was described as part-time (i.e., 24 h/week or less) in 77.8% of homes, including one small (29 residents or less), two medium (30–99), and one large-size home (100–199) in rural and urban areas. The role was described as full-time (35 h/week) in two extra-large homes (>200 beds) located in urban areas. Most participants were the sole interRAI coordinator for their home, with the exception of one large and one extra-large home.

Most participants explained their primary role was to schedule interRAI assessments, gather interRAI data (either by performing assessments themselves or from staff members), enter the data into the system, and interpret the data (alone or with another staff member (e.g., Director of Care [DOC]), who would then communicate the findings to other stakeholders/staff as needed. Participants often were responsible for following up on interRAI reports and considering QI responses. A participant explained the need to appraise and interpret the data stating, “*Together...we look at the data, assess and see what makes sense…Are we coding this wrong? Or is it actually an issue?*” (P9).

The roles and responsibilities varied between homes, and the size of the home, were described as an influencing factor. A participant from a larger home explained, “*We have well over 200 beds so… [I] am busy just doing the assessments…My [Director of Care]…pulled data a lot more, especially because she’ll have meetings with the CEO and the Board*” (P8). Participants described that in smaller homes the interRAI coordinator could not “*delegate [interRAI]…to four or five people*” (P5) and the role may “*only be for a couple hours a week*” (P8).

Previous training

Most participants (*n* = 6) were ‘original interRAI coordinators’ who had been in their role since the interRAI system was first implemented in the province in 2017. These participants had attended an initial 2-day in-person training program organized by the professional association of LTC homes. Participants recalled feeling excited and privileged to be working with an innovative, new technology: 

“*I found it very impressive that we were, you know, it always feels like New Brunswick is behind the times on technology. Here’s something that we’re doing that is actually like with technology, something that we’re actually looking at.*” (P8)

Participants expressed that their initial in-person education was “*very helpful at the time*” (P9) and “*the practice...was crucial to being able to do the work...by myself*” (P2). However, following their initial training participants described feeling unprepared for their role. A participant shared, “*The hardest part was trying to find a way to implement it…We didn’t have any guidance or instruction*” (P2) and “*It felt like we were dropped like a hot potato*” (P9) after the training. All interview participants later completed virtual training regarding the interRAI system and the LTCF assessment form provided by CIHI. Some participants explained that they “*redid*” (P5) the virtual training periodically because changes and updates were not communicated well by CIHI. Only one participant mentioned attending educational webinars offered by CIHI consistently (P1).

Use and implementation of interRAI data

The extent of interRAI data use (beyond assessment and data collection) varied between homes. One participant described being at the beginning stages of using interRAI data in their home, explaining, “*We’re going to be getting into [using the reports]...we haven’t really been following up on the reports because of...all the [paper] charting*” (P07). Most commonly, participants described using the interRAI data to monitor organization trends in QIs. Restraint use and resident falls were the most frequently discussed QIs, while pain, antipsychotic use, depression, and pressure ulcers were also mentioned. Interest in comparing their organization-level data to provincial or national benchmarks varied. One participant shared, “*We work with the data during our falls meetings and restraints...We don’t do a whole lot of comparison to any other facility because...it’s not relevant to the frontline work... where we are is of interest*” (P5). Some participants had advanced to using the interRAI data to implement QI projects, with success: “*We’ve had reduced restraint use, I’m happy to say. That was a big, big deal. That was the first one we kind of tackled*” (P8).

Fewer participants had progressed to using the interRAI data to inform resident care planning and person-centered care. A participant explained, “*It is very helpful when it comes to resident specific care planning and looking at trends in the residents and managing their particular needs. Whether...nurses have the time to look at that kind of management level is a different can of beans altogether*” (P2). A participant described, “*We watch our trends, we see if they’re going up, we see if they’re going down, we try to investigate if there is a change and if there is something new*”(P2). Barriers to advancing the use of interRAI data in participant organizations were described as staff time constraints, limited training, reliance on a single interRAI coordinator, inconsistencies in coding (defining and inputting) assessment data, and having paper-based (rather than electronic) charting.

QI strategies

Participants described using QI strategies to address QIs in their respective homes including accurate coding, staff and family education, use of special equipment, policy changes, and peer outreach. Investigating the accuracy of the assessment data and coding according to the CIHI definitions (particularly for restraints) was commonly the ‘first’ QI strategy described by participants, followed by implementing simple, practical changes (e.g., removing seat belts from residents who no longer tried to stand), followed by staff education about the QI and associated problem. The final QI strategy employed often involved implementing a policy to guide and sustain organizational change. A participant described their QI process and success addressing restraints:

“*We did see a large reduction in restraints when we looked at how we were coding them... [S]ome education and modifications to our policies and procedures...based on the LTCF and interRAI information really made a difference of how we were coding... and it did spark... education of families on [when there is a] restraint...how should we get rid of it?*”(P1)

Participants described strategies used for staff education such as sharing QI reports, creating self-directed learning modules, and facilitating education sessions on indicators (e.g., pain) to promote staff engagement and problem-solving. A participant explained, “*I’ve also used these reports to show the members of the care team...to try to engage them...to make sure the data...is accurate, and that it’s important...It just helps everybody own a piece of this when we start to talk about safety and falls and restraints*” (P9).

Multiple participants described the importance of educating families, particularly around the inappropriate use of restraints and anti-psychotics. A participant explained, “*many families come from hospital environments where restraints are common*” (P8). Special equipment, such as motion sensors and protective gear like wedges, helmets, and hip protectors, were also implemented to improve resident safety and reduce falls (P1, P6). Examples of policy changes included revising a restraint policy and consent form to clearly outline the risks and benefits (P6). Participants also emphasized the importance of reaching out to other LTC homes for guidance and QI ideas. One participant explained that they had visited a smaller home to see how they were using equipment like motion sensors to reduce falls (P1). Collectively, these strategies reflect a comprehensive approach to improving care quality through education, equipment, policy revisions, and inter-home collaboration.

#### 3.3.2. Theme 2: Perceptions of the Intervention

Overall, participants’ perceptions of the intervention were largely positive. Many participants reported that both the education sessions and open-call meetings were beneficial and served different purposes. Key areas of interest included restraints, falls, and pain management. For instance, one participant highlighted, “*restraints was the biggest highlight of everything... falls being the second*” (P1), while another mentioned curiosity around “how you would code tilt chairs” and the challenge of defining a fall consistently (P1). Despite the focus on restraints and falls, some participants noted a relative lack of discussion on pain, which one described as “one of the problems that I want to tackle here in our facility” (P1). Participants appreciated the opportunities for peer support and exchange of ideas, with one stating, “I think those sessions were invaluable... I think they were very valuable” (P9). Although progress varied across facilities, the gradual improvement encouraged by facilitators was acknowledged as beneficial (P6). Overall, the sessions were seen as a valuable resource for improving the use of interRAI data for quality improvement.

Facilitators and barriers

Participants identified several facilitators and barriers to attending and learning from the education intervention. Key facilitators included the accessibility of the virtual format, which allowed participants to “*access that information from wherever they were*” (P1), and the structured Q&A format, where “*questions be distributed*” after the session (P4). Role modeling and collaboration were also perceived as valuable, with one participant noting that “*working in working groups with [Facilitator’s Name] has been the most helpful thing*” (P5). However, barriers to engagement emerged as the sessions progressed, with some describing “*a little bit of dead air*” as momentum waned, likely due to the virtual format (P1). Additionally, missed communication was a challenge, as one participant expressed frustration that information was sent to their CEO and “*didn’t filter down*” to the team, resulting in missed sessions (P8). Despite these barriers, the structured approach and opportunities for collaboration were seen as beneficial.

Perceived benefits

Participants identified several key benefits of the education intervention, including increased knowledge, enhanced collaboration, greater consistency, and improved practice. Many described the sessions as informative, with one participant noting, “*I learned a lot where other nursing homes had coding issues*” (P1), and another expressing a clearer understanding of data use, remarking, “*now we understand what they’re looking for and how to code for it*” (P8). Collaboration and role modeling were also highly valued, with facilitators helping to connect coordinators and share best practices across facilities (P2). The intervention fostered increased consistency in data interpretation and reporting, with participants working toward “*getting everybody on the same page*” (P1). Furthermore, the sessions led to tangible improvements in practice, particularly around restraint use and antipsychotic prescriptions, with one participant stating that the intervention “*absolutely changes how I practice*” (P4). Overall, the education intervention supported both personal and system-level improvements in long-term care settings.

#### 3.3.3. Theme 3: Challenges and Recommendations

Participants recommended several improvements for future education surrounding interRAI and QI projects. A step-wise or staged approach to ongoing training was suggested, with one participant advocating for “*guided updates*” that build on foundational knowledge, progressing to more advanced topics over time (P2). Increased collaboration among interRAI coordinators was another key recommendation, as participants valued the opportunity to “*help people through different struggles*” and share best practices (P1). Engaging LTC staff and leadership teams beyond the interRAI coordinators was seen as essential for broader system improvement, with one participant emphasizing the need to involve others in the learning process (P2). Participants also proposed new QI project ideas, including better monitoring of family quality indicators, stating it would be useful to “*have a little more detail on how these QIs are sorted*” (P8). These suggestions aim to enhance both the depth and reach of future educational interventions.

## 4. Discussion

This study explored the perceptions and experiences of interRAI coordinators in NB LTC homes and evaluated an educational intervention designed to enhance the utilization of interRAI data for quality improvement and begin to build a community of practice among interRAI coordinators. The findings highlight significant variability in the roles and practices of interRAI coordinators across LTC homes and demonstrate the positive impact of the intervention on coordinators’ knowledge, collaboration, and use of interRAI data. However, challenges related to resource constraints, role isolation, and limited integration of interRAI data into clinical care planning were also evident.

The educational intervention was well received by interRAI coordinators, who appreciated its role in fostering peer support, sharing best practices, and enhancing knowledge about data utilization. The structured format of virtual sessions and open-call meetings facilitated collaboration among coordinators and encouraged the exchange of strategies to address quality indicators such as restraint use and resident falls. These findings align with previous research emphasizing the importance of tailored, ongoing education to maximize the effectiveness of interRAI systems in LTC settings [8].

While the intervention successfully addressed foundational knowledge gaps and promoted collaboration, participants emphasized the need for continuous education that builds on their existing knowledge and evolves with the interRAI system. The lack of sustained support following initial interRAI training was a recurring theme, underscoring the necessity of follow-up training and mentorship to empower coordinators in their roles. These findings suggest that a staged, iterative approach to education could further enhance coordinators’ capacity to implement data-informed quality improvement initiatives.

The variability in the roles and responsibilities of interRAI coordinators across LTC homes emerged as a significant finding. Coordinators in smaller homes often faced time constraints and resource limitations, juggling multiple responsibilities with minimal support. In contrast, larger homes with more staff resources allowed coordinators to dedicate greater time to interRAI-related tasks, including data interpretation and quality improvement. This disparity highlights the need for tailored support strategies that account for the unique challenges faced by coordinators in smaller, resource-limited facilities.

Participants frequently expressed a sense of isolation in their roles, particularly in smaller homes where they were the sole coordinators. This finding is consistent with studies that highlight the challenges of implementing data-driven initiatives in under-resourced LTC settings [17]. Collaborative networks or communities of practice, such as those modeled by the SQLI [7,12], could help address this isolation by facilitating knowledge-sharing and peer support among coordinators.

The extent to which interRAI data were utilized varied widely among participants. Most coordinators used the data to monitor organizational trends in quality indicators, such as restraint use and falls, but fewer advanced to using the data for resident-specific care planning. This gap between data collection and application in clinical practice highlights an underutilization of interRAI’s potential to inform person-centered care [18]. Time constraints, limited training, and reliance on a single coordinator were identified as barriers to integrating interRAI data into daily care planning.

Despite these challenges, successful quality improvement initiatives, such as reductions in restraint use, demonstrate the potential impact of interRAI data when effectively utilized. Participants described practical strategies, including accurate data coding, staff education, policy changes, and the use of specialized equipment, to address quality indicators. These findings underscore the importance of providing coordinators with the resources and training needed to transition from data monitoring to actionable QI initiatives.

### 4.1. Implications for Practice

The findings from this study underscore the critical role of interRAI coordinators in enhancing the quality of care in LTC settings across NB. The positive reception of the educational intervention highlights a shared recognition among coordinators of the importance of continuous education and collaboration. However, the variability in coordinators’ roles and their ability to fully leverage interRAI data points to significant gaps in care quality and outcomes across different LTC homes. This suggests the need for tailored educational resources and support structures that address the unique challenges faced by coordinators, particularly in smaller, resource-limited facilities.

Chronic under-resourcing in LTC has hindered coordinators’ ability to engage in broader comparative analyses, revealing a disconnect between organizational priorities and the benefits of utilizing interRAI data for quality improvement. This gap may result in missed opportunities to enhance person-centered care, underscoring the need for policy changes that address staffing shortages and provide sufficient support to coordinators. Ensuring adequate staffing and resources for interRAI coordinators, especially in smaller facilities, will empower them to fully capitalize on data to improve overall care quality.

To address these challenges, future educational interventions should take a staged approach, offering guided updates that build on foundational knowledge while addressing more advanced topics. Engaging a broader range of LTC staff and leadership teams in interRAI education could further enhance the integration of data into clinical practice and organizational decision-making. Collaborative networks and communities of practice could also support coordinators by fostering peer learning and the sharing of best practices, ultimately strengthening the use of interRAI data to drive continuous improvement in care quality across LTC homes.

### 4.2. Implications for Future Research

The central theme of this study is the significant lack of support for interRAI coordinators, directly hampering their ability to fully utilize interRAI data in NB LTC homes. Our findings reveal that coordinators with access to adequate resources and support are more successful in implementing interRAI data for quality improvement and care planning. However, most LTC homes in New Brunswick are smaller, and coordinators in these settings often feel isolated, face barriers such as chronic understaffing and under-resourcing, and struggle to utilize interRAI data effectively. These challenges highlight the urgent need for future research to explore various ways to provide better support to interRAI coordinators.

One key area for future study is conducting longitudinal research on the long-term impact of educational interventions on coordinators, particularly in terms of whether these interventions are effective or if they should be customized to meet the specific needs of coordinators. Additionally, it is important to explore if educational interventions could be supplemented with mentoring programs that connect coordinators, reducing the isolation felt by those in smaller homes, and fostering a community of practice. Research should assess the effectiveness of such programs in enhancing coordinators’ confidence and capacity to implement data-informed initiatives.

Moreover, the disparity in data utilization across homes often stems from the lack of support for coordinators. Given the positive response to collaborative strategies during the educational intervention, such as peer learning and sharing best practices, future research should further investigate how cross-disciplinary teams can support interRAI coordinators. Specifically, exploring how healthcare professionals, such as therapists and administrators, can collaborate with coordinators to enhance the use of interRAI data could improve clinical outcomes and drive quality improvement initiatives across LTC homes.

### 4.3. Limitations

This study has several limitations that should be considered when interpreting the findings. First, the small sample size and reliance on voluntary participation may limit the generalizability of the results to a broader population of interRAI coordinators in LTC homes. Having more participants would strengthen the conclusions, offering a deeper understanding of the educational intervention’s impact. Further, the absence of demographic data on four participants introduces potential bias, as it is unclear whether these participants differ from the others in ways that could influence the findings. Additionally, all interviewed interRAI coordinators perceived the educational intervention as a positive experience overall; however, while most participants attended all or nearly all the education sessions, some had low engagement, potentially impacting the reliability of our findings. Another notable limitation is the lack of longitudinal follow-up data, preventing an assessment of the long-term effects of the educational intervention on coordinators’ practices and the sustained utilization of interRAI data over time. 

## 5. Conclusions

This study highlights the positive impact of an educational intervention on interRAI coordinators’ knowledge, collaboration, and use of interRAI data in NB LTC homes. However, significant variability in coordinator roles and challenges in data utilization underscore the need for increased support, such as ongoing, tailored education. Addressing this lack of support can help maximize the potential of interRAI data to enhance care quality and outcomes for residents. New Brunswick LTC homes possess valuable data that could drive improvements, but coordinators, who are eager to use these data, require more support to do so effectively.

## Figures and Tables

**Table 1 healthcare-12-02592-t001:** Interview participant characteristics (N = 9).

Characteristics		N (%)
Gender	Female	9 (100)
Age in years	31–40	1 (11.1)
	51–60	3 (33.3)
	61–70	1 (11.1)
	Missing ^a^	4 (44.4)
Employment status	Full-time	2 (22.2)
	Part-time	6 (66.7)
	Casual	1 (11.1)
Current role(s)	Staff nurse	2 (22.2)
	interRAI coordinator	9 (100)
Years worked in LTC	1–10	3 (33.3)
	Great than 10	4 (44.4)
	Missing ^a^	2 (22.2)
Years in current role	1–5	3 (33.3)
	6–10	6 (66.7)

^a^ Some participants chose to not provide their age or years working in long-term care, resulting in missing demographic data.

## Data Availability

The data presented in this study are available on request from the corresponding authors.

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
