# Peer review of "Improving Data-Informed Care in New Brunswick Long-Term Care Homes: A Qualitative Study on an Educational Intervention for interRAI Coordinators"

_healthcare, 2024, doi:10.3390/healthcare12242592_

Round 1
Reviewer 1 Report
Comments and Suggestions for Authors
It gives me pleasure to review the manuscript on ‘Improving data-informed care in New Brunswick long-term care homes: A qualitative study on educational intervention for interRAI coordinators’. This manuscript explores the experiences of interRAI coordinators and their perceptions of an educational intervention designed to enhance their ability to use interRAI data for quality care effectively. This research would undoubtedly contribute to the evidence-based informed care model for the aging population.
According to me, the manuscript is well-conceived, articulated, and presented in a scholarly manner. The introduction including the justification of the research has been presented systematically.
In material and methods section gives relevant information related to the educational intervention, recruitment of participants, the approach followed for a qualitative study, and the analysis undertaken are adequately explained.
The discussion section also provides implications and limitations of the research.
There are certain minor comments on my part as given below:
1. Authors need to provide more descriptions of virtual interviews including questions included in the semi-structured interview guide. Only two questions are included here (refer lines 171-178). Details should be provided as a supplementary file.
2. Lines 138-140 - the meaning of the sentence is not clear. While 73 [Blinded] homes were invited to attend the intervention through email invitations to the LTC home administrator or leader. The result section (Line 189) mentions an average of 51 LTC staff participants attended the education sessions and open-call meetings. More clarifications are required here.
3. In Table 2 – what does ‘missing’ mean?
4. Future studies – Is it possible to recommend future studies based on major issues that emerged from the findings? Refer to conclusion (apart from larger, more diverse samples and explore the long-term impact of educational interventions on interRAI data utilization).
Author Response
Thank you for your review. Although you labeled your comments as 'minor', they really stimulated deeper thinking into the meaning of our findings - thank you!!
Comment 1. Authors need to provide more descriptions of virtual interviews including questions included in the semi-structured interview guide. Only two questions are included here (refer lines 171-178). Details should be provided as a supplementary file.
Response 1: Thank you! We had considered adding the whole interview guide in the supplementary materials. It is now included in the supplementary materials and the manuscript line 176 now reads as Virtual interviews were conducted through Microsoft Teams using a semi-structured interview guide with questions such as: “what was surprising about the interRAI data in the sessions?” and “what quality improvements has your home initiated based on interRAI data?” (the full interview guide can be found in the supplementary materials).
Comment 2: Lines 138-140 - the meaning of the sentence is not clear. While 73 [Blinded] homes were invited to attend the intervention through email invitations to the LTC home administrator or leader. The result section (Line 189) mentions an average of 51 LTC staff participants attended the education sessions and open-call meetings. More clarifications are required here.
Response 2: Thank you - that definitely reads as confusing! Clarification added in lines 140-143 as: Invitations to participate in the educational intervention were sent to anglophone and francophone LTC staff responsible for collecting and reporting interRAI data from New Brunswick’s 73 LTC homes through email invitations to the LTC home administrator or leader.
Comment 3: In Table 2 – what does ‘missing’ mean?
Response 3: 'Missing' means some participants did not disclose all requested demographic data. Clarification has been added as a superscript note at the end of the table as follows: a Some participants chose to not provide their age or years working in long-term care, resulting in missing demographic data
Comment 4. Future studies – Is it possible to recommend future studies based on major issues that emerged from the findings? Refer to conclusion (apart from larger, more diverse samples and explore the long-term impact of educational interventions on interRAI data utilization).
Response 4: This is a really, really good point! We have added an "implications for future research" section and altered the limitations accordingly, lines 492-527:
4.2 Implications for Future Research
The central theme of this study is the significant lack of support for interRAI coordinators, directly hampering their ability to utilize interRAI data in NB LTC homes fully. Our findings reveal that coordinators with access to adequate resources and support are more successful in implementing interRAI data for quality improvement and care planning. However, most LTC homes in New Brunswick are smaller, and coordinators in these settings often feel isolated, face barriers such as chronic understaffing and under-resourcing, and struggle to utilize interRAI data effectively. These challenges highlight the urgent need for future research to explore various ways to provide better support to interRAI coordinators.
One key area for future study is the long-term impact of educational interventions on coordinators, particularly in terms of whether these interventions should be customized to meet the specific needs of coordinators. Additionally, it is important to explore if educational interventions could be supplemented with mentoring programs that connect coordinators, reducing the isolation felt by those in smaller homes, and fostering a community of practice. Research should assess the effectiveness of such programs in enhancing coordinators' confidence and capacity to implement data-informed initiatives.
Moreover, the disparity in data utilization across homes often stems from the lack of support for coordinators. Given the positive response to collaborative strategies during the educational intervention, such as peer learning and sharing best practices, future research should further investigate how cross-disciplinary teams can support interRAI coordinators. Specifically, exploring how healthcare professionals, such as therapists and administrators, can collaborate with coordinators to enhance the use of interRAI data could improve clinical outcomes and drive quality improvement initiatives across LTC homes.
4.3. Limitations
This study has several limitations that should be considered when interpreting the findings. First, the small sample size and reliance on voluntary participation may limit the generalizability of the results to a broader population of interRAI coordinators in LTC homes. Additionally, the absence of demographic data on four participants introduces potential bias, as it is unclear whether these participants differ from the others in ways that could influence the findings. Another notable limitation is the lack of longitudinal follow-up data, preventing an assessment of the long-term effects of the educational intervention on coordinators’ practices and the sustained utilization of interRAI data over time.
Reviewer 2 Report
Comments and Suggestions for Authors
Dear Authors, very important work regarding an actual research topic.
I suggest to you some strategies for strenghtening the contribution of your work:
1) Remove the blinding sections in your manuscript, as it is a single blind review. In addition, please adopt the MDPI style for your bibliography;
2) The study relies on a small sample size of nine participants, which may limit the generalizability of the findings across different long-term care settings. Expanding the participant pool could strengthen the conclusions and provide a more comprehensive understanding of the educational intervention's impact. Please, address this issue in the 4.2. Limitations section, expanding the direction of future research.
3) There is a notable variation in the engagement levels of participants in the intervention. Some participants attended all sessions, while others had minimal attendance. This inconsistency may influence the reliability of the results and the overall conclusions about the intervention’s effectiveness. Addressing barriers to participation should be considered.
4) The absence of longitudinal data on the sustained impact of the educational intervention limits the ability to evaluate its long-term effectiveness. Incorporating follow-up studies could help determine whether the initial benefits are maintained over time. Please, consider this suggestion for your future research and specificate it in the manuscript.
Author Response
Thank you for your thorough review!
Comment 1: Remove the blinding sections in your manuscript, as it is a single blind review. In addition, please adopt the MDPI style for your bibliography;
Response 1: Thank you for pointing out this oversight! All blinding has been removed (lines 19, 22, 53, 98, 106, 117, 140, 224, 291, 469, 532, 549). We acknowledge there were errors in the italicizing and bolding of journal names, volumes, and dates within article references in the reference list. These are now fixed as per the reference list highlights
Comment 2: The study relies on a small sample size of nine participants, which may limit the generalizability of the findings across different long-term care settings. Expanding the participant pool could strengthen the conclusions and provide a more comprehensive understanding of the educational intervention's impact. Please, address this issue in the 4.2. Limitations section, expanding the direction of future research.
Response 2: Thank you! The limitations section has been expanded to include this limitation (lines 519-523):
This study has several limitations that should be considered when interpreting the findings. First, the small sample size and reliance on voluntary participation may limit the generalizability of the results to a broader population of interRAI coordinators in LTC homes. Having more participants would strengthen the conclusions, offering a deeper understanding of the educational intervention's impact.
Comment 3: There is a notable variation in the engagement levels of participants in the intervention. Some participants attended all sessions, while others had minimal attendance. This inconsistency may influence the reliability of the results and the overall conclusions about the intervention’s effectiveness. Addressing barriers to participation should be considered.
Response 3: This is an excellent point, thank you! The limitations section has been altered to include this fact (lines 525-529): Additionally, all interviewed interRAI coordinators perceived the educational intervention as a positive experience overall; however, while most participants attended all or nearly all the education sessions, some had low engagement, potentially impacting the reliability of our findings.
Comment 4: The absence of longitudinal data on the sustained impact of the educational intervention limits the ability to evaluate its long-term effectiveness. Incorporating follow-up studies could help determine whether the initial benefits are maintained over time. Please, consider this suggestion for your future research and specificate it in the manuscript.
Response 4: As per another reviewer's comments, a "implications for future research" section has been added, where the absence of longitudinal data is more clearly specified (lines 502-504):
One key area for future study is conducting longitudinal research on the long-term impact of educational interventions on coordinators, particularly in terms of whether these interventions are effective or if they should be customized to meet the specific needs of coordinators
Thank you!